# The Potential Psychological Mechanism of Subjective Well-Being in Migrant Workers: A Structural Equation Models Analysis

**DOI:** 10.3390/ijerph16122229

**Published:** 2019-06-24

**Authors:** Hao Chen, Lei Wang, Yanxia Wei, Bo Ye, Junming Dai, Junling Gao, Fan Wang, Hua Fu

**Affiliations:** 1Department of Preventive Medicine and Health Education, School of Public Health, Fudan University, Shanghai 200032, China; haochen18@fudan.edu.cn (H.C.); leiwang16@fudan.edu.cn (L.W.); clili17@163.com (Y.W.); yb_iyx1314@163.com (B.Y.); jmdai@fudan.edu.cn (J.D.); jlgao@fudan.edu.cn (J.G.); 2Department of Clinical Nursing, School of nursing and health management, Shanghai University of Medicine & Health Sciences, Shanghai 201318, China; 3Department of Politics, School of Humanities and Social Science, East China Normal University, Shanghai 200241, China; wangfan512@126.com

**Keywords:** migrant workers, sense of coherence, generalized resistant resources, well-being

## Abstract

*Objective*: The aim of this study was to identify the potential psychological mechanism of well-being in migrants in Shanghai, China. *Methods*: A cross-sectional study was conducted in 2018. First, a literature review was conducted to understand the salutogenesis of migrants in China. Then, 2573 random participants were recruited from six workplaces and public places in six districts of Shanghai. The Chinese versions of the Patient Health Questionnaire-9 (PHQ-9), the Personal Wellbeing Index (PWI), and the Sense of Coherence (SOC) Scale were used to evaluate the depression, subjective well-being (SWB), and SOC of migrants. The *t*-test, ANOVA, and multiple linear regression and structural equation models (SEM) were used to analyze the correlations and paths among generalized resistant resources (GRRs), SOC, PHQ, and SWB. *Results*: The subjects were aged between 18 and 58 (mean, 28.17; SD, 6.99). SOC showed a positive correlation with SWB (*r* = 0.46, *p* < 0.001) and a negative correlation to PHQ (*r* = −0.53, *p* < 0.001). After controlling for the demographic characteristics, we found that PHQ, SOC (comprehensibility, manageability, meaningfulness), and GRRs (income ratio, marital status) contributed 33.3% of the variance in SWB, and their linear regression coefficients were: −0.32 (*p* < 0.001), 0.09 (*p* < 0.001), 0.09 (*p* < 0.001), 0.15 (*p* < 0.001), 0.06 (*p* < 0.05), and 0.16 (*p* < 0.05), respectively. These findings not only confirmed the direct association among SOC, PHQ and SWB, but also verified two underlying mechanisms regarding the mediating effect of SOC by using SEM: (1) GRRs (income ratio, marital status) are positively associated with a higher SOC, which further contributes to favorable SWB; and (2) PHQ is negatively associated with poor SWB indirectly via SOC. *Conclusion*: Migrant workers with low SOC and high PHQ are vulnerable to poor well-being levels. Meanwhile, GRRs (income ratio, marital status) may strengthen the SOC level, and can be regarded as the basis of intervention. Further investigation may be needed to focus on external psychological support factors.

## 1. Introduction

Since piloting Premier Deng Xiaoping’s reform and open policy in 1978, China has been incorporated into the global financial system, driving rapid economic growth and facilitating the establishment of numerous factories in Special Economic Zones [1], which induced significant migration from rural areas into cities due to job opportunities and favorable incomes. The total number of migrant workers in China was 286.52 million in 2017, representing a 1.7% increase as compared to 2016. Specifically, the number of migrant workers who left their hometowns to work in other places was 171.85 million, representing an increase of 1.15% with respect to 2016 [2]. Migrant workers make a major contribution to Chinese industrialization and urbanization, but they are marginalized in urban communities and have to accept a harsher living environment than local residents. Because they lack household registration where they work, they cannot enjoy certain rights, such as free education and access to social welfare [3]. According to Reports of China National Monitoring Survey of Migrant Workers (2015), the proportion of housing in total consumption of migrant workers was 48.6%; this proportion was only 24.1 % in urban residents, thus indicating a tremendous difference in consumption structure [4]. Furthermore, migrant workers have to pay expensive rent and have no rights available to buy a house even they have enough money because of Hukou (the household registration system) restrictions [5]. The main function of Hukou is to allocate resources as a social control and administrative mechanism, not only in housing but also in education. Children of migrant workers rarely receive abundant and high-quality public education resources in the megacities, and most of them are sent to local privately-operated schools specifically for migrants’ children, where the education quality and environment are not comparable to those of public schools [6]. This policy exclusion of Hukou has also brought economic exclusion in labor rights, including labor contracts, employment benefits, social insurance, delays, and deductions in pay and violence [7]. Besides political and economic exclusion, migrant workers have experienced psychological exclusion in the form of social stigma and discrimination. For example, they are interrogated by the police in public spaces, and are barred from entering service venues more often than local residents [8]. Social stigmas also set obstacles to their participation in activities and organizations [9]. Therefore, under the pressure from various forms of disenfranchisement and disadvantages because of their migrant status, migrant workers may be vulnerable to mental illness and even depression, and this is exacerbated by limited access to essential public social services [10]. According to a random sampling survey of 3031 migrant workers conducted in Shenzhen, China, approximately 34.4% suffered from common mental health problems [11].

Exploring the relationship between adverse social processes and the mental health of migrant workers was a main research focus of previous studies [7]. Most studies have found that the higher work pressure [12] and poor economic status of migrant workers working without a contract or with deferred pay are associated with worse mental health status [13,14]. Beyond the social status of migration, some studies have focused on internal psychological attributes. It had been found that migrant workers who perceive discrimination derived from exceeding disparity compared with wealthier natives or lack of a sense of belonging to the city are more likely to experience worse mental health [15,16]. However, there are also a handful of collective-level studies that have aimed to identify the positive social forces that can benefit migrant workers’ mental health. One study suggested that neighborhood social cohesion is significantly negatively correlated with psychological distress when controlling for individual-level social resources [17]. Lin [14] also found that participating in a high number of activities was beneficial to mental health status.

Although some Chinese studies have focused on positive factors or processes to improve the mental health and well-being of migrants, few studies have explored the potential mechanism to enhance migrants’ psychological situation by using a comprehensive health promotion strategy. Salutogenesis, developed by Aaron Antonovsky in the late 1970s, is an ongoing movement that is concerned with various resources that can improve well-being as well as strengthen a person’s health maintenance [18]. According to Antonovsky, the first critical concept of salutogenesis is a sense of coherence (SOC). This is defined as a general orientation of responding to pervasive and sustained complicated life experiences with (1) the perception of internal and external stimuli as structured, predictable, and explicable; (2) a conviction that one has the available resources to meet the demands of these stimuli; (3) the belief that all of these demands have a reason and are worth meeting. Three components of SOC are comprehensibility, manageability, and meaningfulness. People with a strong sense of coherence believe that their life is comprehensible, manageable, and meaningful, which empowers them to identify resources in their surroundings and then use these resources to regulate their health behaviors [19]. Japanese research has suggested that SOC is positively associated with physical and mental well-being in workers [20,21], and SOC is generally believed to be a mediator of the relationship between stress and mental health status [22]. Albertsen explained the mediating effect of SOC between work environment and stress symptoms in 2053 Danish employees [23]. The second key concept of salutogenesis is generalized resistance resources (GRRs), which provide prerequisites for the development of the SOC and can be found within people as resources tied to their personality and capacity and also to their direct and indirect contexts, such as income, education, physical exercise, and marital status [24]. One study found that SOC was changed considerably by GRRs over a four-year period among a representative sample of the Canadian labor force [25]. Another study determined that marital status, family income, years of formal education, and physical exercise were positively associated with SOC [24].

The aforementioned results of previous research led us to consider how migrants’ well-being could be improved through developing their SOC. However, little is known about the SOC of migrant workers in China. We thus decided to explore the potential psychological mechanism of well-being and whether or how GRRs can strength the SOC among migrant workers in Shanghai. According to the salutogenic model of health and the aforementioned results, several hypotheses were confirmed in the present study: (1) SOC has a positive correlation with subjective well-being (SWB) and a negative correlation with depression; (2) some types of GRRs (marital status, family income, physical exercise and education level) or sociodemographic variables are positively associated with SOC; and (3) SOC is beneficial to well-being and plays a mediating role between depression and well-being in migrant workers.

## 2. Materials and Methods

### 2.1. Study Population and Settings

We conducted a cross-sectional study in Shanghai, China, from July 2018 to September 2018. By the end of 2017, the city had 24.2 million permanent residents, of whom 9.7 million were migrant workers with residence permits [26]. For this study, 2573 migrant workers were recruited through two procedures. Occasional sampling was used to select a total of 471 workers or salespeople from shopping malls, restaurants, barbershops, and other types of stores in six urban districts. The remaining 2120 migrant workers were selected using the cluster sampling method from six large workplaces (≥300 employees) in the suburban districts of Pu Dong, Min Hang, Song Jiang, Qing Pu, and Jia Ding districts of Shanghai. The factories produced electromechanical products, precision electronics, semiconductors, precision molds, computers, cell phones, and costumes. The distribution of sampling sites is shown in Figure 1.

All subjects gave their informed consent for inclusion before they participated in the study. The study was conducted in accordance with the Declaration of Helsinki, and the protocol was approved by the Ethics Committee of IRB#2015-12-0574.

### 2.2. Measures

#### 2.2.1. Sense of Coherence

The sense of coherence was measured by the Chinese version of the SOC Scale (SOC-13), which contains three dimensions: comprehensibility (five items), manageability (four items), and meaningfulness (four items) [27]. Each of the 13 items is scored on a seven-point Likert scale, ranging from 1 (“very often”) to 7 (“never or very seldom”). The total SOC score is obtained by adding the 13 items’ scores after revising the five negatively worded items, with a higher SOC score indicating a greater sense of coherence. The Chinese version of the SOC-13 (C-SOC-13) has been demonstrated to have acceptable reliability and validity [28]. In the present study, the Cronbach’s alpha coefficient for the internal consistency of the C-SOC-13 was 0.81.

#### 2.2.2. Depression

The Patient Health Questionnaire (PHQ-9) is a simple, self-administered version instrument that quantifies the frequency of being bothered by nine evaluative items over the past two weeks [29]. Responders were asked to rate each item on a Likert scale from 0 (“not at all”) to 3 (“nearly every day”), with the items summed for an overall score ranging from 0–27. The questionnaire has been translated into multiple languages and has shown to have good cultural equivalence, including the Chinese version (C-PHQ-9) [30]. With the Diagnostic and Statistical Manual of Mental Disorders, Fifth Edition (DSM-IV) diagnosis of major depression as the reference, the C-PHQ-9 demonstrated good reliability in Chinese adults with a cut-off score of ≥7 (sensitivity, 0.86; specificity, 0.86).

#### 2.2.3. Subjective Well-Being

The Personal Wellbeing Index (PWI), which has been validated in different countries and cultural backgrounds, was selected to measure SWB. The index contains eight different domains: (1) standard of living; (2) personal health; (3) life achievements; (4) personal relationships; (5) personal safety; (6) community connectedness; (7) future security; and (8) religion and spirituality [31]. However, the religion-related items were removed from the SWB, since religion is a sensitive subject in Chinese culture, and less than 10% of the Chinese population is religious [32]. Thus, a validated seven-item Chinese version of the self-reported Well-Being Scale (C-SWB-7) was adopted to measure life satisfaction, with a Cronbach’s *α* value 0.960 [33]. The items were rated on an 11-point Likert scale ranging from 0 (very dissatisfied) to 10 (very satisfied).

#### 2.2.4. Generalized Resistance Resources

Based on The Hitchhiker’s Guide to Salutogenesis [34], which lists 13 different types of GRRs, we selected five GRRs according to the Chinese context: (1) income ratio: Whether the subject’s income was sufficient to cover their household expenses (enough income, balanced income, insufficient income); (2) education attainment (secondary school and below, senior high school, junior college, undergraduate degree, master’s degree and above); (3) marital status (spinsterhood—single, widowed, or divorced; married—married or cohabiting); (4) family accompaniment: Whether family members accompanied the respondents to Shanghai (yes/no); and (5) preventive activity (estimated using part of the Chinese long-form version of the International Physical Activity Questionnaire about moderate activity) [35]. Two questions regarding the frequency and duration of moderate activity in a seven-day week were also asked. In accordance with the current recommendations for the practice of physical activity, this study defined preventive activity as at least 150 min of moderate physical activity per week [36]. All of these questions were answered via self-report in an interview.

#### 2.2.5. Covariates

Covariates in this study included age, gender, occupation (direct labor, administrative staff, clerk, professional, service staff, others), district (urban, suburb), and monthly income (≤3000 Chinese yuan (CNY), 3001–6000 CNY, 6001–9999 CNY, ≥10,000 CNY).

### 2.3. Data Analyses

Data were entered using EpiData 3.1 and analyzed by SPSS 22.0 (SPSS, Chicago, IL, USA). The associations between sociodemographic characteristics and SWB, SOC, and PHQ were expressed as the mean ± Standard Deviation (SD) and assessed using student-t (T) and Analysis of Variance (ANOVA)/ANOVA tests. Multiple linear regression models were used to assess the relationships between SOC, PHQ, and SWB, adjusting for covariates. *p* < 0.05 was considered statistically significant. Structural equation modeling was used to assess the standardized coefficients among SOC, PHQ, and SWB in migrant workers. Amos 22.0 (SPSS Amos, Chicago, IL, USA) was used to determine whether the data fit the model.

## 3. Results

### 3.1. Sample Characteristics and Associations with SWB, SOC, and PHQ

The 2573 migrants were aged between 15 and 58 (mean, 28.24; *SD*, 7.33); 49.8% of them were men, and 18.8% were urban residents. Less than a half (45.5%) had received senior high school or higher education. Over half (51.5%) of workers had a balanced income. The majority of the migrants’ jobs (40.4%) consisted of direct labor. Only 25.3% were accompanied by family, and over half (53.6%) were single. SWB varied with age, sex, income ratio, education, job, family accompaniment, and marital status (*p* < 0.05). According to the PHQ scale cut-off point of 9, the prevalence of depression was 24.3% among the respondents. SOC was found to vary with income ratio, education, district, and marital status (*p* < 0.05). PHQ also varied with age, sex, income ratio, education, job, district, family accompaniment, and marital status (*p* < 0.05). The specific associations between demographic characteristics and SWB, SOC, and PHQ are shown in Table 1.

### 3.2. Correlation of SWB, SOC, and PHQ

Table 2 presents the means, standard deviations, reliability, and bivariate correlations. The mean score on the SWB scale was 48.39 (*SD*, 12.96), the mean score on the SOC was 62.29 (*SD*, 10.64), and the mean score on the PHQ scale was 6.66 (SD, 4.75). All scales showed an acceptable level reliability (>0.80), as evaluated by Cronbach’s *α*. SWB, SOC, and PHQ were all significantly correlated in the expected direction. SWB was moderately positively correlated with the total score of SOC (*r* = 0.469, *p* < 0.001) and negatively correlated with the PHQ score (*r* = −0.469, *p* < 0.001). There was also a negative correlation between PHQ and SOC (*r* = −0.544, *p* < 0.001).

### 3.3. Association among SOC, PHQ, and SWB

The results of the stepwise multiple regression with dummy variables for SWB yielded the outcomes reported in Table 3. Model 0, including only three components of SOC and PHQ, indicated that those three components were all associated with SWB without GRRs and controlling covariates. Model 1 found that participants who were married and had enough income or a balanced income had higher levels of SWB (*β* = 0.04, 0.16, and 0.11, respectively) compared with single participants and those with insufficient income. Furthermore, Model 1 predicted an additional 3% variance over and above the 30.0% of Model 0. After controlling for five sociodemographic characteristics, all of the components of SOC were still positively associated with SWB, while PHQ remained negatively associated with SWB. In the results of Model 2, GRRs showed a similar effect as in Model 1. Finally, the Model 2 contributed 33.3% of the variance to the SWB and did not differ significantly from Model 1. There was no multicollinearity, and the variance inflation factor coefficients were all being below 1.5.

### 3.4. SEM of SWB, SOC, GRR, and PHQ

Structural equation modeling was used to test the hypotheses on the psychological mechanism and confirm our standardized structural coefficients (SSCs) results using maximum likelihood estimation with the covariance matrix. Compared to the criteria of goodness-of-fit statistics, it was a better fit to the data (*χ*^2^/degree of freedom, df = 264.68; Root Mean Square Error of Approximation, RMSEA = 0.08; Comparative Fit Index, CFI = 0.95), and all of the paths were statistically significant (*p <* 0.05). The estimates are shown in Figure 2. SOC was a mediating variable that was influenced in the negative direction by PHQ (SSCs = −0.58, *p <* 0.01) and two types of GRRs (SSCs = 0.06 and 0.03, *p <* 0.05). SOC was also positively and directly associated with SWB (SSCs = 0.30, *p <* 0.01). Meanwhile, PHQ was indirectly and negatively associated with SWB (SSCs = −0.58 × 0.30, *p <* 0.01) and directly and negatively associated with SWB (SSCs = −0.32, *p <* 0.01).

## 4. Discussion

This study showed that 24.3% of 2573 Chinese migrant workers had depression, a higher prevalence than the general population [37]. They also had weaker personal well-being scores (SWB score = 48.3) than China’s urban population (SWB score = 67.1) [38]. These findings indicated poor mental health of migrant workers. Meanwhile, the SOC level of migrants (62.02 points) was similar to that of patients in disadvantaged groups, such as elderly patients with coronary heart disease (62.20 points) who lived in northern China [39] and Hong Kong mothers with autism (60.06 points) [40].

The present findings showed that the risk factors of depression were generally consistent with those from previous studies, including being single status, having an inadequate income, and living in an urban district [11,41,42]. The previous literature indicated that socio-demographics (age and job) and some GRRs (marital status, education, family accompany, regular activity) were associated with well-being [3,43,44,45,46]. The present study not only supported those findings but also found a cost/benefit imbalance to be an important predictor of lower SWB and higher depression. The second-generation younger migrant workers who constitute the main stream of migration do not wish to return to their home towns and sustain extremely high rent and basic living costs, while their poor social skills and lower resilience means they cannot enjoy a high level of entertainment and improve their satisfaction with life [45,47]. Moreover, one study found that “subjective economic status”, related to one’s sense of well-being, had a stronger relationship than objective economic status [48].

Furthermore, the majority of the correlations of SOC with sociodemographic characteristics, depression, and SWB found in our study are consistent with many previous findings from cross-sectional and cohort studies [21,49,50,51]. In this regard, we found that migrant workers who were married, had enough income, or lived in suburbs had higher SOC scores than those who were single, had an inadequate income, or lived in an urban area. Previous findings from a sample of Finnish participants aged 65–69 years suggested that marriage status can promote a family to adopt a healthier lifestyle and behaviors, which can strengthen SOC, especially for men [24]. Another possibility considered by Volane is that people with a poor SOC face obstacles to having close marriage relationships [52]. As far as load-balanced or sufficient income is associated with a better SOC, it is probably the case that migrant workers with sufficient economic status can attain a high quality of life and entertainment more easily than others because those resources may stimulate the sense of life meaning (which is a part of SOC). As mentioned by Geyer [53], SOC is an attitude of people who occupy higher socio-economic positions, but it seems that the concept of SOC needs to be developed further. With the exception of two domains of GRRs (married, enough income), the residence of migrant workers can be regarded as a type of specific resistance resource (SRR), which is defined as a structure (amenity or service) of a site (government or agency) that is activated specifically to avoid (reinterpret or adapt to) a specific stressor [34]. Making a profit entices migrant workers to come to the megacity of Shanghai in high numbers; however, even though they obtain a high salary, they must bear extremely high living costs and cramped living conditions. This contradiction is an urgent and specific stressor for migrants. However, we found that urban migrant workers lived in a more severe environment with a much more expensive living cost than suburban migrant workers. Therefore, living in suburbs may alleviate the specific economic stressors of migrant workers and improve their utilization of GRRs, thereby supporting SOC as an SRR. The present study demonstrated the positive effect of SOC on migrant depression, possibly because migrants with a strong SOC seem to perceive life pressure as more benign and improvable, thus feeling less dispirited and passive as well as coping appropriately with stressors using certain strategies [51]. The present findings also provide confirmatory evidence for the previous results suggesting that an SOC buffers pressure, raising positive well-being and decreasing psychological stressors [54,55].

The present study predicted that SOC would play a mediating role between adverse dilemmas and positive well-being. We also tested a mediation model where GRRs (income ratio, education attainment, marital status, family accompaniment, preventive activity) were assumed to support SOC, which in turn was hypothesized to enhance personal well-being. Pressure is the typical effect of negative experience in salutogenesis theory [18]. However, migration may not only expose migrants to the various extrinsic pressures from their occupation, the economy, and family members, but also induce intrinsic passivity and loneliness without family accompaniment. All of these factors together may induce depression in migrants. We used a depression scale to assess occupational stress and to test our hypotheses by structural equation modeling. The ultimate model fitting result partially supported the hypotheses and the main claim of Antonovsky’s model [56]. The present study suggested that depression not only negatively affected well-being directly, but also did so indirectly via a mediator: SOC. This structural path from depression to well-being was consistent with recent findings [57,58], which means that appropriate intervention strategies to strengthen SOC may help protect against depression and weaken the negative effect on well-being through this path. However, only two GRRs (income ratio, marital status) were demonstrated to have a significant, albeit small, positive sustaining effect on SOC, in turn promoting well-being; these GRRs differed somewhat from the findings of a study focused on elderly Finnish people [24]. Besides marital status, preventive activity was another factor found to promote SOC in the aged population study, while income ratio showed no relationship with SOC. This discrepancy may be explained by the different living demands or goals of these two populations, such as making a profit for younger migrant workers in a megacity and improving the quality of life for older local residents.

## 5. Conclusions

Using a representative sample of migrant workers, this study elucidated the potential of GRRs and the mediating effect of SOC on the relationship between depression and GRRs by establishing pathways to better well-being. Given the poor mental health and low well-being of marginalized migrant workers, although more job opportunities have been created, there remain regions where more attention should be paid to promoting the mental health and well-being of migrants. Based on the results of the present study, approaches such as fostering social bonds and developing activities aimed at promoting marital relations and social welfare could be particularly promising for improving migrant worker well-being.

## 6. Limitation

A causal relationship could not be found in this cross-sectional study, so it is uncertain whether GRRs shape the SOC or whether a strong SOC facilitates the use of GRRs. Several important GRRs, including religion or beliefs and adverse events (e.g., a serious accident or lawsuit experience), were excluded due to their sensitivity in Chinese culture.

## Figures and Tables

**Figure 1 ijerph-16-02229-f001:**
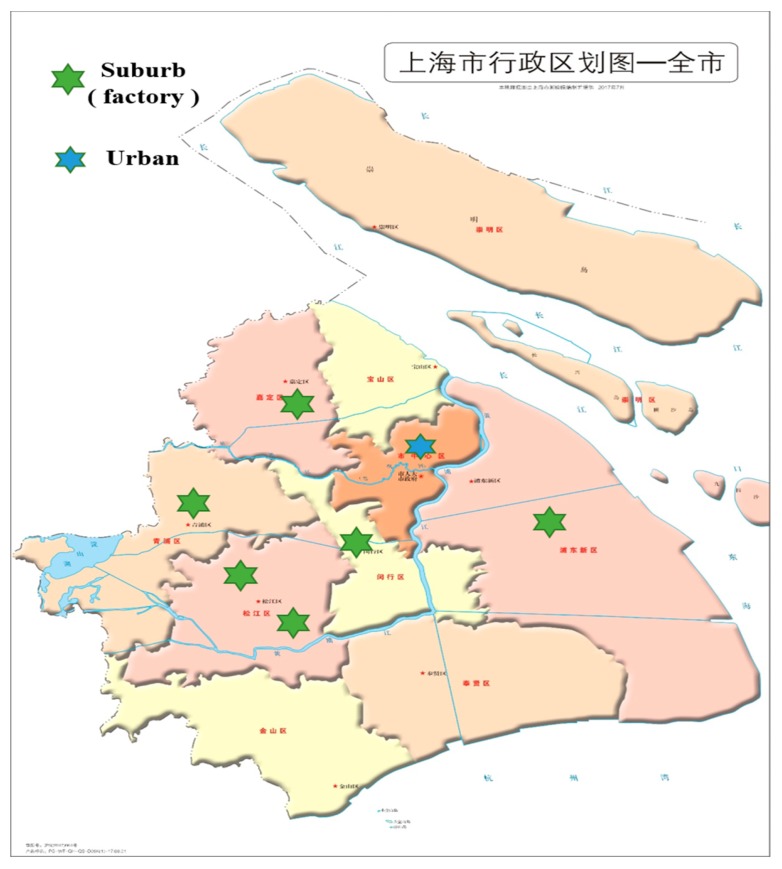
The distribution of sampling sites.

**Figure 2 ijerph-16-02229-f002:**
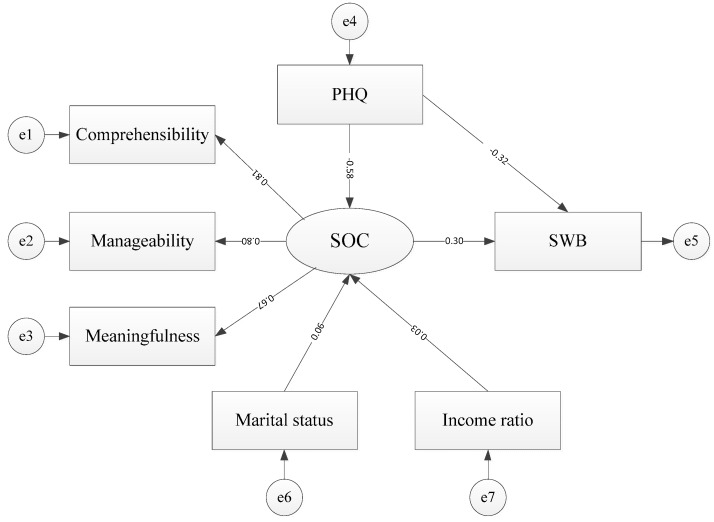
The paths among PHQ, SOC, GRRs, and SWB. (e1–e7: residuals of observed variables)

**Table 1 ijerph-16-02229-t001:** The specific associations among demographic characteristics and SWB, SOC, and PHQ.

Variate	*n* (%)	SWB		SOC		PHQ	
		Mean (SD)	T/F, *p*	Mean (SD)	T/F, *p*	Mean (SD)	T/F, *p*
Gender							
Male	1281 (49.79)	49.60 (13.17)	4.90	63.07 (11.13)	4.90	6.45 (4.98)	2.35
Female	1292 (50.21)	47.11 (12.62)	<0.001	60.98 (10.57)	<0.001	6.89 (4.50)	0.019
Age	-	-					
18–24	906 (35.21)	48.06 (13.02)	3.15	61.09 (10.84)	4.04	6.82 (4.75)	3.96
25–39	1276 (49.59)	48.01 (12.49)	0.024	62.59 (10.69)	0.007	6.96 (4.67)	0.008
40–49	280 (10.88)	49.80 (13.49)	-	62.76 (11.41)	-	6.66 (4.89)	-
50–59	111 (4.32)	50.98 (15.60)	-	61.19 (11.97)	-	5.18 (5.02)	-
Monthly income							
≤3000 CNY	335 (13.02)	47.52 (14.19)	1.38	60.51 (10.96)	3.52	7.13 (4.86)	1.69
3001–6000 CNY	1572 (61.10)	48.57 (12.99)	0.25	62.36 (11.06)	0.14	6.52 (4.76)	0.17
6001–9999 CNY	513 (19.94)	47.85 (12.21)	-	61.61 (10.40)	-	6.79 (4.64)	-
≥10,000 CNY	153 (5.94)	49.65 (12.04)	-	63.23 (10.45)	-	6.76 (4.71)	-
District							
Urban	475 (18.46)	47.54 (13.38)	2.30	59.32 (10.66)	36.23	7.99 (5.08)	46.38
Suburb	2098 (81.54)	48.54 (12.85)	0.13	62.63 (10.86)	<0.001	6.37 (4.62)	<0.001
Jobs							
Direct labor	1043 (40.54)	48.62 (13.49)	1.73	62.74 (11.14)	5.45	6.16 (4.85)	9.45
Administrative staff	119 (4.62)	49.14 (12.86)	0.12	62.76 (11.06)	<0.001	6.59 (4.32)	<0.001
Clerk	305 (11.85)	48.81 (12.98)	-	61.70 (10.06)	-	6.59 (4.06)	-
Professional	406 (15.78)	49.01 (12.25)	-	63.01 (10.67)	-	6.49 (4.68)	-
Service staff	482 (18.73)	46.88 (12.71)	-	59.94 (10.61)	-	7.92 (4.95)	-
Others	218 (8.48)	48.03 (11.99)	-	61.38 (11.24)	-	6.82 (4.60)	-
Education							
Secondary school and below	585 (22.74)	49.90 (13.44)	6.16 <0.001	62.11 (12.19)	6.57 <0.001	6.12 (5.14)	5.40 <0.001
Senior high school	789 (30.66)	48.74 (13.14)	-	63.25 (10.61)	-	6.55 (4.80)	-
Junior college	576 (22.39)	47.89 (13.20)	-	61.66 (10.12)	-	6.81 (4.48)	-
Undergraduate College and above	598 (24.21)	46.84 (11.79)		60.72 (10.51)		7.19 (4.47)	
Income ratio							
Enough income	342 (13.29)	53.84 (11.01)	78.10	64.15 (11.04)	16.27	5.47 (4.55)	44.99
Balanced income	1327 (51.57)	49.49 (12.59)	<0.001	62.48 (10.57)	<0.001	6.20 (4.46)	<0.001
Insufficient income	904 (35.14)	44.60 (13.11)		60.54 (11.13)	-	7.81 (5.00)	-
Family accompany							
Yes	651 (25.30)	49.43 (12.58)	6.05	62.60 (11.04)	2.50	6.36 (4.51)	3.70
No	1922 (74.70)	47.99 (13.06)	0.014	61.82 (10.84)	0.114	6.77 (4.82)	0.054
Marital status							
Spinsterhood	1378 (53.56)	47.45 (13.16)	14.42	61.23 (10.78)	15.59	6.94 (4.87)	9.94
Married	1195 (46.44)	49.39 (12.63)	<0.001	62.93 (10.97)	<0.001	6.35 (4.59)	0.002
Preventive activity							
Yes	211 (8.20)	50.55 (13.72)	6.61	63.88 (10.80)	6.73	6.42 (5.60)	0.62
No	2362 (91.80)	48.16 (12.87)	0.01	61.85 (10.89)	0.01	6.69 (4.67)	0.433

T—statistic of student-*t* test; F—statistic of ANOVA test.

**Table 2 ijerph-16-02229-t002:** The descriptive statistics reliabilities and Pearson’s correlation of SWB, SOC, and PHQ.

		Mean	SD	Cronbach’s α	SWB	SOC
1	SWB	48.35	12.95	0.92		
2	SOC	62.02	10.90	0.81	0.46 ***	
3	PHQ	6.67	4.75	0.88	−0.49 ***	−0.53 ***

* *p* < 0.05; ** *p* < 0.01; *** *p* < 0.01.

**Table 3 ijerph-16-02229-t003:** Associations among SWB, PHQ, and generalized resistant resources (GRRs).

Variate	Association with Case-Level Personal Well-Being (β, 95% CI)		
	Model 0	Model 1	Model 2
SOC	-	-	-
Comprehensibility	0.10 (0.13; 0.38) ***	0.08 (0.10; 0.34) ***	0.09 (0.10; 0.36) ***
Manageability	0.09 (0.13; 0.41) ***	0.09 (0.15; 0.42) ***	0.09 (0.13; 0.40) ***
Meaningfulness	0.15 (0.37; 0.66) ***	0.15 (0.37; 0.65) ***	0.15 (0.36; 0.65) ***
PHQ	−0.35 (−1.05; −0.84) ***	−0.32 (−0.97; −0.76) ***	−0.32 (−0.98; −0.77) ***
GRRs	-	-	-
Marital status			
Spinsterhood	-	Reference	Reference
Married	-	0.04 (0.02; 2.85) *	0.06 (0.45; 2.70) *
Income ratio			
Enough income	-	0.16 (4.84; 7.54) *	0.16 (4.74; 7.50) *
Balanced income	-	0.11 (1.99; 3.81) *	0.11 (2.01; 3.86) *
Insufficient income	-	Reference	Reference
Education attainment			
Secondary school and below	-	Reference	Reference
Senior high school	-	−0.04 (−2.16; 0.13)	−0.04 (−2.12; 0.46)
Junior college	-	−0.03 (−2.09; 0.17)	−0.03 (−2.12; 0.46)
Undergraduate College and above	-	−0.06 (−2.89; −0.47)	−0.04 (−2.12; 0.46)
Family accompany			
Yes	-	Reference	Reference
No	-	−0.02 (0.27; −1.64)	−0.03 (−1.83; 0.28)
Preventive activity			
Yes	-	0.03 (−0.25; 2.77)	0.02 (−0.47; 2.57)
No	-	Reference	Reference
*R* ^2^	0.30	0.33	0.33

Controlled variables of Model 2: age, gender, occupation, district and monthly income; * *p* < 0.05; ** *p* < 0.01; *** *p* < 0.01; Confidence Interval, CI.

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
