# Peer review of "The Potential Psychological Mechanism of Subjective Well-Being in Migrant Workers: A Structural Equation Models Analysis"

_ijerph, 2019, doi:10.3390/ijerph16122229_

Round 1

Reviewer 1 Report

I really appreciated your effort to improve your manuscript. At the end of the review process, I've just one more recommendation: regarding your hypotheses, there is an incoherence between the introduction section and the presentation of the results. I think that the way you describe your results, and consequently the discussion, should be made by taking into account the hypotheses order (in the introduction section), or vice versa. 

Probably, as you reported in the results section, the correct way is to introduce first the hypothesis regarding the correlations, then the background variables, and finally the mediation effect. Pay attention to that progressive exposition of the hypotheses in the abstract as well.

Author Response

Dear Reviewer:

Thank you for your comments concerning our manuscript entitled “Potential Psychological Mechanism of Subjective Well-Being in Migrant Workers: A Structural Equation Models Analysis”. Those comments are so valuable and very helpful for revising and improving critical structure of our manuscript. We have studied comments carefully and have made correction which we hope meet with approval. Revised portion are marked in colorful by using “Track Changes” function of Microsoft Word.

We are followed your advice and revised three parts:

1. We adjusted the order of hypothesis keeping to your suggestions that “first correlations, then the background variables, and finally the mediation effect”. (Line 112-117)

2. Correspondingly, the order of result in abstract was also adjusted and rewritten. To be specific, the result of correlation and regression analysis was corresponding to first hypothesis (Line 22-27); the positive association between SOC and GRRs was corresponding to second hypothesis (Line 29-31); the mediating effect of SOC was corresponding to third hypothesis (Line 30-31).

3. We also moved two evidences from last paragraph of introduction (Line 107-110) to the above paragraph. Because the majority contents of the above paragraph were the evidences regarding the relationship among SOC, well-being and mental health. Those evidences all support the hypothesis. 

Special thanks to you for your good comments. 

Yours sincerely,

Hao Chen

Name: Hua Fu

E-mail: hfu@fudan.edu.cn

Reviewer 2 Report

The changes that have been made addressed the points raised in my initial review. 

Author Response

Dear Reviewer:

Thank you for your comments concerning our manuscript entitled “Potential Psychological Mechanism of Subjective Well-Being in Migrant Workers: A Structural Equation Models Analysis”.

Initial comments are so valuable and very helpful for revising and improving critical contents of our manuscriptas well as the important guiding significance to our researches. We are horned that our correction had meet with approval.

Special thanks to you for your good comments. 

Yours sincerely,

Hao Chen

Name: Hua Fu

E-mail: hfu@fudan.edu.cn

This manuscript is a resubmission of an earlier submission. The following is a list of the peer review reports and author responses from that submission.

Round 1

Reviewer 1 Report

At the end of your introduction you said: “how migrants’ well-being 84 could be improved through developing their SOC. However, little is known about the SOC of migrant 85 workers in China. We thus decided to explore the potential psychological mechanism of well-being 86 and whether or how GRRs can strength the SOC among migrant workers in Shanghai”

It seems a general aim of your study which could be better transform in one or more hypotheses. In my opinion, your methodology and strategy of analysis could lead you to extract hypotheses from your general aims.

In this regard:

firstly, the interaction effect between gender, age and, sociodemographic variables, on one hand, and SOC and SWB would be an interesting perspective of analysis, even though previous studies have been found inconsistent results;

secondly, you proposed a paths model among PHQ, SOC, GRRs, SWB, which needs for an hypothesis behind it. See for example your sentence in the Discussion section “The present study predicted that a sense of coherence would play a mediating role between 259 adverse dilemma and positive well-being”. Why you didn't think to  expect that mediation effect in the introduction paragraph and consequently in a hypothesis section?

Author Response

Point 1It seems a general aim of your study which could be better transform in one or more hypotheses. In my opinion, your methodology and strategy of analysis could lead you to extract hypotheses from your general aims. Firstly, the interaction effect between gender, age and, sociodemographic variables, on one hand, and SOC and SWB would be an interesting perspective of analysis, even though previous studies have been found inconsistent results.

Response 1 Thanks for your reading my manuscript earnestly and giving such critical opinion of my structure. I had ignored that this hypothesis could be divided into more than one part. Therefore, I followed your advice that extracted my general aim for several parts. 1) It may be found differences in SOC and SWB between different levels of gender, age and sociodemographic variables.2) SOC was beneficial to well-being and played a mediating effect to wellbeing in migrant workers. 3) There existed some type of GRRs associated with SOC(line 90-95).

Point 2Secondly, you proposed a paths model among PHQ, SOC, GRRs, SWB, which needs for an hypothesis behind it. See for example your sentence in the Discussion section “The present study predicted that a sense of coherence would play a mediating role between 259 adverse dilemma and positive well-being”. Why you didn't think to expect that mediation effect in the introduction paragraph and consequently in a hypothesis section?

Response 2 Thanks again for pointing logical and structure deficiency in hypothesis. Therefore, I followed your advice and afford evidence for paths road among PHQ, SOC, GRRs and SWB in introductionline 87-90. However, there is few evidence focus on the mediating effect between depression and SOC. Migration may not only expose migrants to the various extrinsic pressures from occupation, economy, and family, but also induce intrinsic passivity and loneliness without family accompaniment. All of those factors together may induce depression in migrants. We used a depression scale to assess occupational stress and to test our hypotheses by structural equation modeling. (Line 275-278).

Reviewer 2 Report

Thank you for the opportunity to read your work.

The methodology and analysis of data is outlined in an appropriate fashion

This is the report of an interesting piece of work with marginalized migrant workers and their mental health needs. The research has been contextualised but I wonder if a wider readership might benefit from some more background information about the social, economic and political status of migrant workers. The study raises important questions about the impact on mental health of these marginalised groups. This is clearly an issue that is not limited to Shangai or other cities in China but has echoes in megacities across the globe.

Author Response

Point 1:This is the report of an interesting piece of work with marginalized migrant workers and their mental health needs. The research has been contextualized but I wonder if a wider readership might benefit from some more background information about the social, economic and political status of migrant workers. The study raises important questions about the impact on mental health of these marginalized groups. This is clearly an issue that is not limited to Shanghai or other cities in China but has echoes in megacities across the globe.

Response 1: Thanks for your reading my manuscript earnestly and giving such critical opinion. The difference of migrant works’ background would affect the mental health as you mentioned. We thus conducted this survey not only mental health but their economic status, including salary and income ratio. Interestingly, the mental health was associated with perceived income ratio rather than salary (line 180, table 1). In other words, if migrant workers get high salary, they may not get better mental health. Inversely, if migrant workers get not high salary but they perceived enough income, they were more tend to be good mental health. The recessive policy disadvantages are that they lack household registration in their job location, they cannot enjoy certain rights, such as free education and access to social welfare (line 45-46) .Unfortunately, we had not collected political and religion information for sensitivity in Chinese culture. Most of Chinese, especially these marginalized groups, were dislike to tell these information because of privacy they believed. But we would like to explore the difference and generality between the migrant workers from various countries who lived in shanghai in future based on diverse culture context. I hope our research could contribute more echoes for megacities across the globe in the future.
